Cumulative oxygen deficit is a novel predictor for the timing of invasive mechanical ventilation in COVID-19 patients with respiratory distress

Ge Huiqing 1
Zhou Jian-cang 2
Lv FangFang 3
Zhang Junli 3
Yi Jun 4
Yang Changming 5
Zhang Lingwei 6
Zhou Yuhan 6
Ren Binbin 7
Pan Qing 6 pqpq@zjut.edu.cn
http://orcid.org/0000-0002-2336-5323 Zhang Zhongheng 8 zh_zhang1984@zju.edu.cn
1 Department of Respiratory Care, Sir Run Run Shaw Hospital, Zhejiang University School of Medicine , Hangzhou , China
2 Department of Critical Care Medicine, Sir Run Run Shaw Hospital, Zhejiang University School of Medicine , Hangzhou , China
3 Department of Infectious Diseases, Sir Run Run Shaw Hospital, Zhejiang University School of Medicine , Hangzhou , China
4 Thoracic Cardiovascular Surgery, Jingmen First People’s Hospital , Hubei , China
5 Department of Anesthesiology, The First People’s of Hospital of Jingmen City , Hubei , China
6 College of Information Engineering, Zhejiang University of Technology , Hangzhou , China
7 Department of Infectious Disease, Jinhua Municipal Central Hospiltal, Affiliated Jinhua Hospital, Zhejiang University School of Medicine , Jinhua, Zhejiang , China
8 Department of Emergency Medicine, Sir Run Run Shaw Hospital, Zhejiang University School of Medicine , Hangzhou , China
Bauman Eric
Electronic publication date: 2020 Nov 27
Publication date: 2020
Volume: 8
Electronic Location ID: e10497
Received 2020 Sep 20; Accepted 2020 Nov 14
Copyright: © 2020 Ge et al.
Copyright year: 2020
Copyright holder: Ge et al.
License: This is an open access article distributed under the terms of the Creative Commons Attribution License, which permits unrestricted use, distribution, reproduction and adaptation in any medium and for any purpose provided that it is properly attributed. For attribution, the original author(s), title, publication source (PeerJ) and either DOI or URL of the article must be cited.
License URL: https://creativecommons.org/licenses/by/4.0/

Keywords: COVID-19, Mechanical ventilation, Intubation, Oxygenation

Funding: Public Research Project of Emergency Prevention and Treatment of COVID-19 of Jinhua City 2020XG-06 Major Science and Technology Project of Zhejiang Province Health Commission WKJ-ZJ-2112 Binbin Ren received funding from the Public Research Project of Emergency Prevention and Treatment of COVID-19 of Jinhua City (2020XG-06); Jian-cang Zhou received funding from Major Science and Technology Project of Zhejiang Province Health Commission (WKJ-ZJ-2112). The funders had no role in study design, data collection and analysis, decision to publish, or preparation of the manuscript.

==============================
Background and objectives

The timing of invasive mechanical ventilation (IMV) is controversial in COVID-19 patients with acute respiratory hypoxemia. The study aimed to develop a novel predictor called cumulative oxygen deficit (COD) for the risk stratification.

Methods

The study was conducted in four designated hospitals for treating COVID-19 patients in Jingmen, Wuhan, from January to March 2020. COD was defined to account for both the magnitude and duration of hypoxemia. A higher value of COD indicated more oxygen deficit. The predictive performance of COD was calculated in multivariable Cox regression models.

Results

A number of 111 patients including 80 in the non-IMV group and 31 in the IMV group were included. Patients with IMV had substantially lower PaO2 (62 (49, 89) vs. 90.5 (68, 125.25) mmHg; p < 0.001), and higher COD (−6.87 (−29.36, 52.38) vs. −231.68 (−1040.78, 119.83) mmHg·day) than patients without IMV. As compared to patients with COD < 0, patients with COD > 30 mmHg·day had higher risk of fatality (HR: 3.79, 95% CI [2.57–16.93]; p = 0.037), and those with COD > 50 mmHg·day were 10 times more likely to die (HR: 10.45, 95% CI [1.28–85.37]; p = 0.029).

Conclusions

The study developed a novel predictor COD which considered both magnitude and duration of hypoxemia, to assist risk stratification of COVID-19 patients with acute respiratory distress.

Introduction

Coronavirus disease 2019 (COVID-19) has spread all over the world since its first outbreak in Wuhan, China in December 2019 (Wang et al., 2020; Novel Coronavirus Pneumonia Emergency Response Epidemiology Team, 2020). The fatality rate was reported to be around 5% all over the world (Phua et al., 2020). A substantial number of patients (19%) infected with the severe acute respiratory syndrome coronavirus 2 (SARS-CoV-2) will develop respiratory distress and acute lung injury (Wu & McGoogan, 2020; Ruan et al., 2020). Respiratory support becomes important for this type of severe patients (Yang et al., 2020). The surviving sepsis guideline of critically ill COVID-19 patients recommended use of oxygen supplementation to maintain pulse oximetry >90%, followed by non-invasive mechanical ventilation (NIV), high-flow nasal canula (HFNC), invasive mechanical ventilation (IMV) and extracorporeal membrane oxygenation (ECMO). However, there is no specific recommendations for the timing of transition from non-invasive support to IMV, and the recommendations are largely based on expert opinions. For example, the guideline recommends “close monitoring for worsening of respiratory status, and early intubation in a controlled setting if worsening occurs” (Alhazzani et al., 2020). This recommendation is based on best practice statement and there is no data on when IMV should be initiated. In clinical practice, the judgement of “worsening” is subjective and varied substantially between different institutions and physicians. The timing of initiation of IMV is not standardized and is mainly determined by subjective judgement. On the one hand, IMV is able to reverse catastrophic hypoxemia and maintain tissue oxygenation, which is life-saving for COVID-19 patients with severe hypoxemia. On the other hand, IMV can cause ventilator-induced lung injury (Herasevich et al., 2011; Cressoni et al., 2016), and patients on IMV usually require large dose of sedatives, analgesics and even neuromuscular blockades (Jakob et al., 2012; Bellani et al., 2016; Chang et al., 2020). These drugs have significant adverse effects (Barr et al., 2013; Murray et al., 2016). Thus, it is difficult to determine the appropriate timing of IMV.

In our experience, we proposed that the timing of transition from non-invasive oxygenation to IMV should consider both the magnitude of hypoxemia and the duration of the hypoxemia. Thus, we developed a novel marker called Cumulative Oxygen Deficit (COD) to reflect both dimensions. By using a single predictor, we reduce a two-dimension feature to a one-dimension parameter that is comparable among different patients. In our study, we hypothesized that the COD before IMV could be a better biomarker than PaO2 to predict survival outcome.

Methods

Study design and setting

The study was conducted in four designated hospitals for treating COVID-19 patients in Jingmen, Wuhan, from January to March 2020. Medical records were retrospectively reviewed to identify eligible patients and variables. Laboratory tests and type of ventilation support were recorded as longitudinal data. The study was designed as a longitudinal study that all patients were followed until hospital discharge or death. One subject contributed several observation units. Patients were divided into groups with IMV and without IMV during hospitalization. The study was approved by the ethics committee of the First People’s hospital of Jingmen (Approval number: 202002007) and the ethics committee of Sir Run Run Shaw Hospital (20200407-32). Individual patient data were de-identified before analysis. Informed consent was waived as determined by the IRB due to retrospective nature of the study design in accordance to the local regulations.

Study population

COVID-19 was confirmed by either (1) genetic sequencing showed highly homogenous sequence with the known novel coronavirus; or (2) novel coronavirus nucleic acid was positive as confirmed by real time (RT)-PCT in respiratory or blood specimen (Jin et al., 2020; Alhazzani et al., 2020). All patients with respiratory distress with one of the following criteria were eligible: respiratory rate >30/min, or oxygen saturation ≤93%, or PaO2/FiO2 ratio ≤300 mmHg. We screened medical records on admission and identified patients with pulse oximetry ≤93% on room air and requires oxygen therapy (OT). Exclusion criteria included: (1) patients with chronic obstructive pulmonary disease with baseline pulse oximetry <92%; (2) pregnant women; (3) subjects younger than 18 years old; (4) patients with do-not-resuscitate order; and (5) patients with comorbidities such as severe burn, recent major stroke with paralysis, terminally ill malignancy, immunodeficiency and dialysis-dependent renal failure.

Clinical variables

Demographics such as age and sex were recorded. Comorbidities were recorded in broad categories such as those involving respiratory system and cardiovascular system. The smoking history were extracted from the medical records. All laboratory variables were recorded in a longitudinal manner. These included serum lactate, arterial partial oxygen pressure (PaO2), arterial partial pressure of carbon dioxide (PaCO2), base excess (BE), pH, C-reactive protein (CRP), Lymphocyte count, and fraction of inspired oxygenation (FiO2) were extracted.

Respiratory support included OT, NIV, HFNC, IMV and ECMO. The transition time from one type to another was recorded to create a number of time intervals at which a subject was on a specific type of respiratory support. Laboratory variables were then matched to each time interval by their respective measurement time. This created a dataset of counting process that included the start time and end time for an interval.

Clinical outcomes included vital status at hospital discharge, length of stay in the hospital were recorded.

Calculation of cumulative oxygen deficit

For patients with IMV, COD was calculated before the use of IMV. Figure 1 is a sample patient used to illustrate the calculation of COD: COD(mmHg⋅day)=80×(t5−t1)−∑i=14⁡(xi+1+xi)⋅(ti+1−ti)/2, where xi is the value of PaO2 measured in mmHg, and ti is the time at which xi is measured. The low end value of PaO2 was 80 mmHg in our hospital and this value is also physiologically reasonable that the oxygen saturation will not continue to rise above this reference value (Collins et al., 2015). Thus, the COD accounted for both magnitude and duration of hypoxemia before IMV. We hypothesized that the longer a patient was on hypoxemia before IMV, the worse of the survival outcome. On the other hand, the outcome would be not so bad if hypoxemia was immediately corrected with IMV even if the magnitude of hypoxemia is large.

Figure 1 Schematic illustration of the calculation of cumulative oxygen deficit (COD).

The COD was calculated as the difference of the areas under the reference curve and the PaO2-day curve (the light green area in the figure).

Statistical analysis

Demographic and laboratory data were compared between patients with and without IMV. Quantitative data were first tested for normality by using the Kolmogorov–Smirnov (K–S) normality test. Normal data were expressed as mean and standard deviation and were compared between groups with t test. Non-normally distributed data were expressed as median and interquartile range (IQR) and were compared with Wilcoxon Rank Sum test. Categorial variables were expressed as the number and percentage and were compared using chi-square or Fisher’s exact test if appropriate (Zhang et al., 2017).

Alluvial plot was employed to visualize how patients transitioned from different types of respiratory support over time. In patients with IMV, we created multivariable Cox regression model to explore the independent predictors of survival outcome. The COD was categorized into four categories at cutoff values of 0, 30 and 50 mmHg ∙ day. A COD value of 30 mmHg ∙ day is equivalent to 60 mmHg for 1.5 days, and a negative value indicates no oxygen deficit. Other variables such as time from admission to IMV, PaO2, PaCO2, Lactate, lymphocyte count, CRP and BE were adjusted for in the model. These variables were included in multivariable regression model because they were considered to be confounders by domain expertise and/or univariate analysis with p < 0.2. The predictive performance of COD was compared with PaO2 before intubation and the time from admission to intubation. We reported time-dependent AUC for the discriminations from day 7 to 28 after hospital admission (Kamarudin, Cox & Kolamunnage-Dona, 2017).

All statistical analyses were performed using RStudio (Version 1.1.463; R version: 4.0.0).

Results

Study population

A total of 111 patients met the inclusion criteria and were included for analysis. No patients were excluded due to COPD, pregnancy CVA and paralysis. There was no patient being excluded from the participating hospitals. There were 80 patients who did not need IMV, and 31 patients required IMV during hospitalization. Patients with IMV had substantially lower PaO2 (62 (49, 89) vs. 90.5 (68, 125.25) mmHg; p < 0.001), higher pH (7.44 (7.38, 7.47) vs. 7.40 (7.35, 7.43); p = 0.006), higher serum lactate (2.5 (1.7, 3.1) vs. 1.7 (1.1, 2.85) mmol/L, p < 0.036) and higher COD (−6.87 (−29.36, 52.38) vs. −231.68 (−1040.78, 119.83) mmHg ∙ day) than patients without IMV during hospitalization (Table 1). These variables were reported as the first value during hospitalization. The time courses of the transition from different types of respiratory support are shown in Fig. 2. It is noted that larger proportion of patients required IMV in the non-survivors.

Table 1 Comparison between patients with and without invasive mechanical ventilation.

Variables	Total (n = 111)	No IMV (n = 80)	IMV (n = 31)	p	
Age, Median (IQR)	57 (45.5, 68.5)	56 (40, 67.25)	64 (54.5, 70.5)	0.076	
Sex, male (%)	60 (55)	39 (50)	21 (68)	0.143	
Comorbidities					
Respiratory system, n (%)	18 (16)	13 (16)	5 (16)	1	
Cardiovascular system, n (%)	41 (37)	29 (36)	12 (39)	0.983	
Smoking history, n (%)	2 (2)	1 (1)	1 (3)	0.482	
PaCO2 (mmHg), Median (IQR)	43 (39, 47.5)	44 (41, 48)	39 (36, 45)	0.051	
PaO2 (mmHg), Median (IQR)	82 (60, 120)	90.5 (68, 125.25)	62 (49, 89)	<0.001	
pH, Median (IQR)	7.42 (7.35, 7.44)	7.40 (7.35, 7.43)	7.44 (7.38, 7.47)	0.006	
BE (mmol/l), Median (IQR)	2.4 (−2.05, 5.1)	2.35 (−2.57, 5.1)	2.4 (0.2, 5.2)	0.342	
Lactate (mmol/l), Median (IQR)	1.9 (1.3, 3)	1.7 (1.1, 2.85)	2.5 (1.7, 3.1)	0.036	
PF (mmHg), Median (IQR)	224.32 (106.14, 348.02)	250.05 (194.14, 363.64)	111.25 (71.98, 200.27)	<0.001	
CRP (mg/dl), Median (IQR)	8.9 (2.75, 53.2)	5.5 (1.78, 35.83)	19.5 (9.85, 78.05)	<0.001	
Lymphocyte count (mmHg·day), Median (IQR)	0.78 (0.28, 0.86)	0.78 (0.43, 0.89)	0.43 (0.28, 0.66)	0.005	
COD (mmHg·day), Median (IQR)	−50.38 (-827.91, 72.67)	−231.68 (−1040.78, 119.83)	−6.87 (−29.36, 52.38)	0.018	
COD, n (%)				<0.001	
<0 (mmHg·day)	56 (50)	49 (61)	7 (23)		
0–30 (×109/L)	14 (13)	3 (4)	11 (35)		
30–50 (mmHg·day)	9 (8)	5 (6)	4 (13)		
>50 (mmHg·day)	32 (29)	23 (29)	9 (29)		
Hospital LOS (days), Mean ± SD	26.55 ± 11.78	26.1 ± 11.35	27.7 ± 12.97	0.548	
Mortality, n (%)	22 (20)	13 (16)	9 (29)	0.211	
Note:

LOS, length of stay; SD, standard deviation; COD, cumulative oxygen deficit; CRP, C-reactive protein; IQR, interquartile range; BE, base excess.

Figure 2 Alluvial plot showing the transitions of respiratory supports over time.

(A) Survivors. (B) Non-survivors.

Independent association of COD and survival outcome in IMV patients

Cumulative Oxygen Deficit was independently associated with survival outcome in multivariable Cox regression model. As compared to patients with COD < 0, patients with COD from 0 to 30 mmHg⋅day were not more likely to die, whereas those with COD > 30 mmHg⋅day had higher risk of fatality (HR: 3.79, 95% CI [2.57–16.93]; p = 0.037), and those with COD > 50 mmHg⋅day were 10 times more likely to die (HR: 10.45, 95% CI [1.28–85.37]; p = 0.029). The time from admission to intubation, PaO2 and lymphocyte count were not associated with survival outcome (Table 2). The time-dependent AUCs of COD, PaO2 and the time from admission to intubation are shown in Fig. 3. It showed that COD had consistently higher AUCs from day 14 to 21. In other words, COD was the best predictor after day 14. Table 3 shows factors associated with IMV.

Table 2 Multivariable Cox regression model to explore the independent association of cumulative oxygen deficit and hazard of death in patients with invasive mechanical ventilation.

	HR [95% CI]	p	
COD (< 0 as reference)	1		
0–30	1.45 [0.98–3.47]	0.052	
30–50	3.79 [2.57–16.93]	0.037	
>50	10.45 [1.28–85.37]	0.029	
Time from admission to intubation	0.78 [0.59–1.03]	0.078	
Age	1.20 [1.00–1.44]	0.056	
Sex (female as reference)	9.84 [0.43–225.59]	0.152	
PaCO2	0.85 [0.60–1.19]	0.344	
PaO2	1.00 [0.97–1.03]	0.939	
pH	0.70 [0.44–1.13]	0.144	
BE	1.89 [0.70–5.08]	0.206	
CRP	0.98 [0.94–1.01]	0.233	
Lymphocyte count	2.21 [0.04–115.48]	0.695	
Note:

COD, cumulative oxygen deficit; CRP, C-reactive protein; IQR, interquartile range; BE, base excess; HR, hazard ratio; CI, confidence interval.

Figure 3 Time-dependent AUCs for cumulative oxygen deficit, PaO2 and the time from admission to intubation.

The AUC of cumulative oxygen deficit was significantly higher than the other two indices from day 14 to 24.

Table 3 Factors associated with use of IMV.

Variables	HR [95% CI]	p	
PaO2 (for every 10-mmHg increase)	0.91 [0.84–0.99]	0.022	
PaCO2	1.02 [0.98–1.06]	0.337	
Age	1.02 [0.99–1.04]	0.235	
pH (for every 0.01 increase)	1.02 [0.96–1.09]	0.565	
CRP	1.00 [0.99–1.01]	0.824	
Lymphocyte count	0.27 [0.09–0.81]	0.020	
Lactate	1.56 [1.17–2.08]	0.003	
Note:

CRP, C-reactive protein; HR, hazard ratio; CI, confidence interval.

Discussion

The study developed a novel biomarker COD which considered both magnitude and duration of hypoxemia, to assist the timing of IMV in patients with COVID-19. In patients with IMV during hospitalization, COD before intubation was a strong predictor of survival outcome. Patients with COD > 30 mmHg⋅day, which is equivalent to a persistent hypoxemia with PaO2 of 60 mmHg for 1.5 days, are more likely to die during hospitalization. Patients in crowded hospital during COVID-19 pandemic were more likely to experience this situation. The time dependent AUCs of COD were substantially higher than that of the PaO2 or the time from admission to intubation alone. Clinical implication of this finding is that we need to consider both the magnitude and duration of hypoxemia before IMV is considered. Long duration of mild hypoxemia, which is prevalent in clinical practice under NIV, may be dangerous for COVID-19 patients.

Many studies have been conducted to address the question on whether NIV should be used for patients with pulmonary/direct ARDS, but the results are conflicting (Chawla et al., 2020). NIV was not associated with improved mortality or length of stay, compared with patients who were intubated without trying NIV in a cohort of Middle East Respiratory Syndrome (MERS) patients. Furthermore, most patients (92.4%) who had tried NIV were eventually managed with IMV (Alraddadi et al., 2019). However, this was a retrospective study and the initiation of IMV was not standardized prospectively. Our study indicated that large COD can be harmful and the correction of COD with IMV might be beneficial. This could be explained by potential adverse effects of NIV including large tidal volumes and injurious transpulmonary pressures (Brochard et al., 2014). These adverse effects of NIV could be avoided by using IMV. For example, protective ventilation strategy can be performed with IMV (Zhang et al., 2015; Fan, Brodie & Slutsky, 2018), but it is impossible under NIV. Furthermore, the use of NIV or HFNC can delay IMV, leading to emergency or more unstable intubations (Brochard, 2003). Thus, IMV should be considered as early as possible if the COD reaches 30 mmHg⋅day, without trying NIV or HFNC to delay intubation.

Arterial partial oxygen pressure and its derivatives such as PF ratio are well established risk factor for mortality outcome in patients with ARDS. Thus, PF ratio is used to classify ARDS patients into mild, moderate and severe cases (ARDS Definition Task Force et al., 2012). However, this risk classification system considers only the magnitude of hypoxemia (Cartotto et al., 2016; Dai et al., 2019). Our results suggest that the duration of hypoxemia can be equally important. A strength of our study was that all measurements of PaO2 were collected longitudinally, allowing for the calculation of the area under the PaO2-day curve to derive a novel predictor. Our analysis focused on patients with IMV and found that the predictive performance for survival outcome of COD before intubation was substantially better than PaO2 or the time from admission to intubation. The latter two indices are the two components of COD. The combination of the two indices substantially improves the predictive discrimination for mechanically ventilated patients. Although direct causal inference that the use of IMV to reduce COD can improve survival outcome cannot be established with current analysis, our result identified a modifiable risk factor for survival outcome. It is reasonable to deduce that reducing COD as early as possible with IMV can be beneficial.

Several limitations should be acknowledged in the study. First, the study was retrospective in design, and many unmeasured confounders may exist to influence the choice of respiratory supports (Uddin et al., 2016). The presence of such unmeasured confounders will compromise the effectiveness of the propensity score matching procedure. Second, the use of NIV or HFNC was completely at the discretion of the treating physician. There was no standard protocol in participating hospitals. Thus, it is difficult to determine whether the use of NIV or HFNC could benefit COVID-19 induced ARDS. Third, for patients without IMV, we calculated the COD across all days of hospitalization. This could be biased because the time-dimension was longer than the IMV group. However, since non-IMV group generally did not have oxygen deficit across hospital stay, the COD was substantially lower than the IMV group. Finally, we only included broad categories of comorbidity burden in our analysis (i.e., respiratory system, cardiovascular system), because the retrospective design of the study did not allow detailed information for the calculation of the Elixhauser’s comorbidity index. It is well known that Elixhauser’s comorbidity index is a good quantity for risk stratification of hospitalized patients (Elixhauser et al., 1998). However, this index is designed to work with ICD-9-CM codes in administrative database, which is not applicable to data collected in retrospective studies.

Conclusions

In conclusion, the study developed a novel predictor COD, which considered both magnitude and duration of hypoxemia, to assist the timing of IMV in patients with COVID-19. We suggest IMV should be the preferred ventilatory support once the COD reaches 30 mmHg⋅day, as mortality increases beyond this value.

Supplemental Information

Supplemental Information 1 Dataset.

Click here for additional data file.

Additional Information and Declarations

Competing Interests

Author Contributions

Human Ethics

Data Availability

The authors declare that they have no competing interests.

Huiqing Ge conceived and designed the experiments, authored or reviewed drafts of the paper, and approved the final draft.

Jian-cang Zhou conceived and designed the experiments, prepared figures and/or tables, and approved the final draft.

FangFang Lv conceived and designed the experiments, performed the experiments, prepared figures and/or tables, and approved the final draft.

Junli Zhang conceived and designed the experiments, analyzed the data, prepared figures and/or tables, authored or reviewed drafts of the paper, and approved the final draft.

Jun Yi performed the experiments, analyzed the data, authored or reviewed drafts of the paper, and approved the final draft.

Changming Yang analyzed the data, prepared figures and/or tables, authored or reviewed drafts of the paper, and approved the final draft.

Lingwei Zhang conceived and designed the experiments, performed the experiments, analyzed the data, authored or reviewed drafts of the paper, and approved the final draft.

Yuhan Zhou performed the experiments, analyzed the data, authored or reviewed drafts of the paper, and approved the final draft.

Binbin Ren analyzed the data, authored or reviewed drafts of the paper, and approved the final draft.

Qing Pan performed the experiments, analyzed the data, prepared figures and/or tables, and approved the final draft.

Zhongheng Zhang conceived and designed the experiments, analyzed the data, prepared figures and/or tables, and approved the final draft.

The following information was supplied relating to ethical approvals (i.e., approving body and any reference numbers):

The study was approved by the ethics committee of the First People’s hospital of Jingmen (Approval number: 202002007) and the ethics committee of Sir Run Run Shaw hospital (20200407-32).

The following information was supplied regarding data availability:

Data used in the current analysis is available as a Supplemental File.

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
