# Peer review of "Cumulative oxygen deficit is a novel predictor for the timing of invasive mechanical ventilation in COVID-19 patients with respiratory distress"

_PeerJ, doi:10.7717/peerj.10497_

## Round 0.1 · original submission · Major Revisions

Thank you for your submission. Your manuscript has been reviewed by both a clinical expert and bio-statistical expert. While your manuscript was found to be relevant and timely, we believe carefully following the reviewers recommendations will greatly increase the likelihood of publication.

Also is it possible that you can provide an English language translation of the ethics documentation, as your manuscript is submitted in English for publication in English and while many reviewers are fluent in more than one language, many are not fluent in Chinese. Also to this end, you may find it helpful to use an native language editor as part of your iterative review process.

·

Basic reporting

First of all, I must commend the authors for being able to use the data acquired during a, as I image very difficult time, to try and improve the care for patients.

The basic language is indeed clear and ambiguous. Nevertheless, I would advise the authors to let a native speaker review the text, as there are some issues with omitted 'a's and 'an's and other minor errors. I made some remarks on these in the text, but I imagine I missed a few.

The references are sufficient in number and relevance, and provide sufficient context.

The structure is professional in structure and figures. The raw data is shared.

Experimental design

The research falls within the aims and scope of the journal.

The research question is well defined, absolutely relevant and meaningful. The 'gap' or possible advance due to the research is indeed stated in the article.

The investigation is performed with high technical standard. There could be a certain concern with the ethical standard. In the Netherlands, a waiver has to have been even in case of retrospective research. It is stated in the article that this is not necessary in China. I can't check this. Moreover, the review boards approval is in Chinese, so I can't confirm it is indeed an approval. But I trust the authors on this.

The methods described are lacking some details on certain points. I put remarks in the document where this is the case.

Validity of the findings

The conclusions are twofold.
About the (apparent) main conclusion, about COD, this is certainly linked to the original research question. Moreover, it is based on the results.

The secondary conclusion, is a bit more doubtful. The use of propensity score matching has some drawbacks. I can't completely interpret the statistical steps followed (mostly due to my own lack of technical knowledge in this field), but I'm not completely convinced that the statistical steps are acceptable. A review by a statistician is probably warranted.

More importantly: the second conclusion, IMV is probably (but not 'proven' due to not reaching statistically isignificance) useful for respiratory distress, seems a bit obvious. The proof that patients would have been better off with IMV can't be substantiated. After all, in the midst of a wave of Covid cases, it is very possible that there was overcrowding. So perhaps there was a lack of resources. So it is very possible that people were not put on IMV due to this, and not clinical conclusions that they did not need it.
I would advice to address this in the discussion.

But even more importantly: in my view, the part of propensity score matching actually undermines the biggest conclusion, namely, the use of COD as a 'biomarker', or trigger to start IMV. After all, the end conclusion in the article now, is basically that COD can be used as a predictor of mortality (if above the trigger value). Not so much that the trigger value is a reliable trigger to optimise the start of IMV.

Additional comments

I think it is a relevant article, and worth publishing.
But, like stated above, I think the article would profit from a distinction into the COD (perhaps a bit more about the background for the specifics chosen, advice on standardising time intervals for calculating the COD, etcetera) and the propensity score matching to try and prove the usefulness of IMV in COVID related ARDS. This last part actually weakens the concept of COD in my view.

Reviewer 2 ·

Basic reporting

Thank yo very much for taking the time to do the research on this timely topic and preparing an above average paper. The paper is somewhat clear and relatively free from spelling and grammatical errors; however, could benefit from editing from a native language speaker. There are some corrections needed such as data are plural, datum is singular. Please consider including a sentence or two describing the alluvium plot and a reference. Reference are current.

Experimental design

The authors discuss comorbidities, but in this reviewer's opinion the inclusion of Elixhauser's comorbidity index would substantially improve this work. [The comorbidity index can be used to do this (https://cran.r-project.org/web/packages/comorbidity/comorbidity.pdf).]

Please state how normality was tested and include mean(SD) and median(IQR) for all variables.

Please refer to "Skewed (non-normal) data " as "Non-normally distributed data"

The "rank-sum" tests is correctly referred to as the "Wilcoxon Rank Sum test".

"Chi-square " should not be capitalized

On what basis were variables included for adjustment?

Please state the version of R, not R Studio used for analysis.

Based on the ASA's stand on statistical significance, please do not dichotomize statistical significance (see Wasserstein, Schirm, & Lazar, 2019)

Validity of the findings

The validity of the findings of the study appropriate for the study as are the conclusions. However the manuscript would be strengthened by the inclusion of Elixhauser's comorbidity index and perhaps even a subanalysis using the Charleson-Deyo comorbidity index.

Additional comments

None.

---

## Round 0.2 · Minor Revisions

Thank you for your comprehensive and careful attention to detail as it relates to your recent revision. Please see the very favorable reviews and recommendations from the two reviewers, paying careful attention to Reviewer #2.

·

Basic reporting

No further comments

Experimental design

No further comments

Validity of the findings

No further comments

Additional comments

I found two typos (I think).
Overal my compliments on the swift reply to the reviews!

Reviewer 2 ·

Basic reporting

Thank you for addressing all this reviewer's comments. This version is a much improved manuscript.

Experimental design

Comments addressed

Validity of the findings

Comments addressed

Additional comments

Although not needed for this manuscript please consider, in the future, testing for normality using multiple methods (including a normal probability plot) for triangulation.

In line 52, please replace the word "significantly" with "substantially" of indicate "clinically significant" (otherwise it is assumed you are referring to statistical significance). Alternatively, please use verbiage simular to "statistical evidence of a difference".

In line 173, please replace the word "significantly" with "substantially" of indicate "clinically significant" (otherwise it is assumed you are referring to statistical significance). Alternatively, please use verbiage simular to "statistical evidence of a difference".

In line 199, please replace the word "significantly" with "substantially" of indicate "clinically significant" (otherwise it is assumed you are referring to statistical significance). Alternatively, please use verbiage simular to "statistical evidence of a difference".

In line 226 & 227, please replace the word "significantly" with "substantially" of indicate "clinically significant" (otherwise it is assumed you are referring to statistical significance). Alternatively, please use verbiage simular to "statistical evidence of a difference".

In line 242, please replace the word "significantly" with "substantially" of indicate "clinically significant" (otherwise it is assumed you are referring to statistical significance). Alternatively, please use verbiage simular to "statistical evidence of a difference".

In line 261, please replace the word "significantly" with "substantially" of indicate "clinically significant" (otherwise it is assumed you are referring to statistical significance). Alternatively, please use verbiage simular to "statistical evidence of a difference".

---

## Round 0.3 · accepted · Accept

Thank you for your your continued patience as it relates to the review and revision process of your manuscript. I believe that your revisions have improved the the clarity of your timely and important study.